# Breeding of Modern Rose Cultivars Decreases the Content of Important Biochemical Compounds in Rose Hips

**DOI:** 10.3390/plants12213734

**Published:** 2023-10-31

**Authors:** Nina Kunc, Metka Hudina, Maja Mikulic-Petkovsek, Gregor Osterc

**Affiliations:** Department of Agronomy, Biotechnical Faculty, University of Ljubljana, Jamnikarjeva 101, 1000 Ljubljana, Slovenia; metka.hudina@bf.uni-lj.si (M.H.); maja.mikulic-petkovsek@bf.uni-lj.si (M.M.-P.); gregor.osterc@bf.uni-lj.si (G.O.)

**Keywords:** *Rosa pendulina*, *Rosa spinosissima*, *Rosa gallica*, HPLC, extraction, organic acids, ascorbic acid, carotenoids

## Abstract

This study aimed to determine the content and composition of bioactive compounds in autochthonous rose hips (*R. pendulina*, *R. spinosissima*, and *R. gallica*) and to compare them with the content of bioactive compounds in some cultivars (‘Harstad’, ‘Bourgogne’, ‘Mount Everest’, ‘Poppius’, ‘Fruhlingsduft’, ‘Single Cherry’, ‘Fruhlingsmorgen’, ‘Violacea’, and ‘Splendens’) derived from these main species. Due to insufficient information on how bioactive compound content changes when crossing roses, this study also sought to ascertain whether modern rose hip cultivars are still a sufficiently rich source of bioactive compounds and could, therefore, be potentially used as a functional food. All material was collected in the Arboretum Volčji Potok (Slovenia). The ascorbic acid content was highest in the ‘Harstad’ cultivar (12.79 g/kg FW), and the total organic acid content varied from 1.57 g/kg FW (*R. spinosissima*) to 34.39 g/kg FW (‘Harstad’). Of all the carotenoids analyzed, only lycopene and *β*-carotene were present in all the samples. The total carotenoid content was highest in the ‘Fruhlingsmorgen’ cultivar (100.84 mg/kg FW), derived from *R. spinosissima*, and lowest in the main species, *R. spinosissima* (9.26 mg/kg FW). It can be concluded, therefore, that the content of bioactive compounds in rose hips of modern cultivars is generally lower than in rose hips of old cultivars and original species included in this study. The research results confirm that modern breeding strategies are mainly focused on goals such as abundant flowering and resistance to diseases and pests and not so much on the content of bioactive compounds.

## 1. Introduction

Today, an increasing amount of products called functional foods, nutraceuticals, medicinal, or nutritious foods can be found on the market. These are common products consumed as part of the diet and have proven beneficial effects on the human body. Because of these properties, their demand is rapidly increasing [1]. Jones [2] reports that the emergence of the functional foods industry is due to consumer awareness of a potentially beneficial role of nutrition in managing disease risk, increasing regulatory awareness of the public health benefits of functional foods, and the economic potential of these products. These foods include rose hips, which contain substantial amounts of bioactive compounds that have been shown to have a positive impact on the human body [3,4,5,6,7].

The role of vitamin C or ascorbic acid (1,4-lactone-2,3-dihydrogluconic acid) in plants is to chase free radicals due to their ability to donate reduction equivalents and due to the relative stability of the monodehydroascorbate radical obtained [8]. Its crucial function in plants is to maintain the oxidative status in the active site of several enzymes and to eliminate reactive oxygen species (ROS) generated during photosynthesis [9]. Hoyle [10] reports that scientists at the College of Exeter and the Shimane College in Japan have demonstrated that ascorbic acid is essential for plant growth. They have identified the enzyme GDP-L-galactose phosphorylase, which produces vitamin C or ascorbate in plants. Prior to this research, ascorbate had already been known as an antioxidant helping plants overcome drought stress and the negative effects of UV radiation; however, it had not been known that they cannot grow without it. On the other hand, ascorbic acid plays a significant role in the human body because it cannot be synthesized endogenously and is, therefore, an essential component of people’s diet [11]. Additionally, the lack of ascorbic acid in people’s diets is not only a problem in poor countries but is becoming an increasing problem in developing countries as well, especially in the northern and eastern parts of Europe. Consequently, breeding strategies have been increasingly developed in recent years to breed roses with a higher ascorbic acid content in rose hips. In the last decades of the 20th and the first years of the 21st centuries, breeding programs in roses were mostly focused on good flowering characteristics (large and well-developed flowers, interesting colors, and smell, …) and on pest- and disease-resistant varieties. The breeding of roses with useful rose hips was scarce, yet the purpose is now changing. Green strategies include awareness of the importance of breeding plants with great biochemical potential, and roses with their rose hips could play a key role here.

Breeding programs should also focus on genotype selection through classical breeding, bioengineering, and changes in agronomic conditions to allow the plant to increase ascorbic acid synthesis, somewhat as part of a defense response [12,13]. Kunc et al. [14] reported ascorbic acid content in the hips of the *R. pendulina* species, autochthonous in Slovenia. However, they found that the predominant acid in this species is malic acid. Large amounts of ascorbic acid in the rose hips of *Rosa* spp. have already been described by Darlington [15], Uggla et al. [16], and Babis and Kucharska [17]. Javanmard et al. [18] reported ascorbic acid in five different ecotypes of *R. canina*. Kaushik et al. [19] reported that breeding programs are mostly focused on yield improvement, disease resistance, tolerance to abiotic stresses, and longer lifespan rather than on an increase in the content of bioactive substances. However, there is an ever-augmenting consumer demand for food that is rich in bioactive substances, maintains human health, and prevents diseases at the same time. In the conducted study, they focused on the fact that the combination of conventional and modern breeding strategies would enable the development of a new generation of vegetable varieties with an increased content of phenolic compounds. They concluded that it is crucial to know the candidate genes that would be included in the synthesis of phenolic compounds and that genome editing also creates new opportunities for the creation of new varieties with an increased phenolic content through a non-transgenic approach. Khanizadeh et al. [20] also reported the importance of bioactive substances through crossbreeding by focusing on breeding programs of fruit. They found that the antioxidant composition of the fruit varies between cultivars, with genetics playing a significant role. They mentioned that in the last few years, fruit breeding programs have been directed toward the range of genotypes rich in antioxidant phenolic compounds. As an example, they cited new selected lines of apples and strawberries which differed significantly from the reference varieties in terms of the antioxidant capacity and the total content of phenolic compounds. These lines not only had good disease resistance but also a higher content of bioactive compounds.

The role of organic acids in plants is to regulate osmotic pressure, pH homeostasis, stress resistance, and fruit quality [21,22,23]. They are intermediates of the major carbon metabolism in plant cells and are involved in biochemical pathways, such as glycolysis, the tricarboxylic acid cycle, photorespiration, the glyoxylate cycle, or the C4 photosynthetic cycle [24]. Drincovich et al. [24] demonstrated that organic acids also contribute to controlling the physiological processes of whole plant cells. In general, organic acids preserve the nutritional value of food, prolong their usability, and improve their flavor. Citric acid, malic acid, fumaric acid, and tartaric acid can be used as acidity regulators to change or maintain the pH of food [25]. Qamar and Braunwald [25] reported that hypertension is an increasingly common cause of death; it is, therefore, vital to reduce the salt content in food in such a way that the resulting salty taste remains the same, meaning that the reduction in the salt content does not affect the quality of the food. Wang and Zhang [26] found that malic acid added to low-sodium salts reduced the bitterness caused by the addition of potassium salts and improved the salty taste. That rose hips could potentially be a rich source of organic acids was reported by Cunja et al. [27], who studied the rose hips from *R. canina* and reported high levels of fumaric acid and shikimic acid. Kunc et al. [14] reported the opposite: low levels of fumaric acid and shikimic acid in *R. pendulina*, indicating significant differences in acid levels among rose species. Demir et al. [28] described the levels of ascorbic acid, citric acid, and malic acid in the rose hips of *R. canina*, *R. dumalis*, *R. gallica*, *R. dumalis* subsp. boissieri, and *R. hirtissima* grown in Turkey.

Carotenoids act as membrane stabilizers in plants and hold two main functions in photosynthesis: energy absorption for use in photosynthesis and chlorophyll protection from excessive light damage [29,30]. They function as an additional pigment for the extraction of light by absorbing light in the blue and violet parts of the spectrum (from 400–500 nm), as absorption using chlorophyll is relatively poor in this part of the spectrum [29,30]. They play a major role in protecting the photosynthetic apparatus. As antioxidant compounds, they prevent the photooxidation of chlorophyll in the reaction centers in order to avoid the formation of singlet oxygen or its harmful effects. Singlet oxygen belongs to a group of ROS, which cause damage to membrane lipids, nucleic acids, and proteins. Carotenoids, as singlet scavengers, are more exposed to stress than chlorophylls. Plants adapt to stressful conditions by increasing the intensity of carotenoid synthesis [30,31,32]. They are potent antioxidant and anti-inflammatory micronutrients. *β*-carotene, *α*-carotene, lycopene, lutein, and zeaxanthin are the best-known carotenoids in rose hips that have beneficial effects on human health. Andersson et al. [33] reported that rose hips are a rich source of carotenoids, and that with a suitable harvest time and correct selection of varieties, the carotenoid content in rose hips can be optimized for different purposes. Olsson et al. [34] reported that rose hips have a higher carotenoid content than berries. Hornero-Méndez and Mínguez-Mosquera [35] reported that they identified six major carotenoids (*β*-carotene, lycopene, rubixanthin, gazaniaxanthin, *β*-cryptoxanthin, and zeaxanthin) in the rose hips of *R. mosqueta*, *R. rubiginosa*, and *R. eglanteria*, along with other minor carotenoids (violaxanthin, antheraxanthin, and *γ*-carotene).

The aim of this study was to determine the content of various bioactive compounds, such as ascorbic acid, organic acids, and carotenoids in the rose hips of different modern rose cultivars, ‘Harstad’, ‘Bourgogne’, ‘Mount Everest’, ‘Popius’, ‘Fruhlingsduft’, ‘Single Cherry’, ‘Fruhlingsmorgen’, ‘Violacea’, and ‘Splendens’, as well as in the rose hips of the main rose species from which these cultivars originate—all being native to Slovenia—*R. pendulina*, *R. spinosissima,* and *R. gallica*. This study was conducted on different types of rose hips that grew under the same conditions. Little is known about how the bioactive compounds are transferred to the progeny. Due to different objectives of the crosses in the last decades aimed at increasing the quality of their rose hips (beautiful flowering, high resistance to pests, etc.), it is believed that modern cultivars have mainly lost a high content of various bioactive compounds. Since there is insufficient information on how the content of bioactive compounds changes when rose hips are crossed, this study also aimed to determine whether modern rose hip cultivars are still a sufficiently rich source of bioactive compounds and could, therefore, potentially be used as functional foods.

## 2. Results

Table 1 shows that the rose hips of the ‘Harstad’ cultivar (12.79 g/kg FW) and the rose hips of *R. pendulina* (12.36 g/kg FW) were the richest in ascorbic acid. The lowest content of ascorbic acid was determined in the hip samples of the ‘Single Cherry’ cultivar from the *R. spinosissima* group. In general, ascorbic acid levels in the hips of each cultivar varied widely among cultivars in the same group. The values measured in the hips of the ‘Splendens’ cultivar, originating from the *R. gallica* group, were statistically different from those of the ‘Violacea’ cultivar, originating from the same group. The rose hips of the ‘Poppius’ cultivar were significantly richer in ascorbic acid compared to the other cultivars from the *R. spinosissima* group.

Figure 1 shows the cluster dendrogram for the ascorbic acid content in pulp with skin for all the samples analyzed. Depending on the ascorbic acid content, the samples are divided into three groups. The first group includes the ‘Harstad’ and *R. pendulina* genotypes. The second group includes the ‘Mount Everest’ (*R. pendulina* group), ‘Poppius’ (*R. spinosissima* group), and ‘Splendens’ (*R. gallica* group) genotypes. The third group includes the remaining rose hips: ‘*R. spinosissima*’, ‘*R. gallica*’, ‘Single Cherry’ (*R. spinosissima* group), ‘Bourgogne’ (*R. pendulina* group), ‘Fruhlingsduft’ (*R. spinosissima* group), ‘Fruhlingsmorgen’ (*R. spinosissima* group), and ‘Violacea’ (*R. gallica* group). Very few cultivars were classified within the same group as the main species from which these cultivars were obtained.

The content of organic acids in the rose hips of different cultivars also varied greatly between cultivars of the same group and also with respect to the species from which they were obtained (Table 2). The acid with the highest concentration in the rose hips was quinic, with *R. spinosissima* having an extremely low content and *R. pendulina* and its ‘Harstad’ cultivar having the highest content of quinic acid. The content of citric acid was the highest in the ‘Fruhlingsmorgen’ cultivar among all the examined rose hips; it was significantly higher than the content in the rose hips of *R. spinosissima*, from which the mentioned cultivar is derived. The citric acid content of the cultivars derived from *R. gallica* was lower than the content in the rose hips of *R. gallica* itself. In the cultivars derived from *R. spinosissima* and *R. pendulina*, the citric acid content was higher than in the main species. The malic acid content was lowest in *R. spinosissima* and highest in the samples of the ‘Fruhlingsmorgen’ cultivar from the *R. spinosissima* group. The content of malic acid in the rose hips of all cultivars was higher than in the hips of the original rose plant, except for the ‘Violacea’ cultivar where a slightly lower content was observed than in the original rose hips from *R. gallica*. The content of shikimic acid was lowest in the rose hips of the ‘Violacea’ cultivar and highest in the rose hips of the ‘Fruhlingsmorgen’ cultivar. Fumaric acid was present in the lowest proportions among all acids; its content did not differ significantly between the samples studied. No statistically significant difference was observed in the total content of organic acids.

According to the total organic acid content (Figure 2), the samples are divided into five groups. The first group includes ‘Harstad’ (*R. pendulina* group), the second group includes ‘Fruhlingsmorgen’ (*R. spinosissima* group), *R. pendulina*, and ‘Poppius’ (*R. spinosissima* group). ‘Mount Everest’ (*R. pendulina* group), *R. gallica*, ‘Violacea’ (*R. gallica* group), ‘Bourgogne’ (*R. pendulina* group), and ‘Splendens’ (*R. gallica* group) are representatives of the third group. The fourth group consists of *R. spinosissima* only, and the last group includes ‘Single Cherry’ (*R. spinosissima* group) and ‘Fruhlingsduft’ (*R. spinosissima* group).

Table 3 shows the carotenoids determined in the rose hip samples studied. *β*-carotene was present in the highest proportions in all the samples. The lowest and the highest measured content was found in two cultivars derived from *R. spinosissima*: ‘Fruhlingsduft’ and ‘Fruhlingsmorgen’. The content of *α*-carotene was highest in *R. pendulina* and its cultivars. Lycopene content was lowest in *R. spinosissima* and highest in ‘Harstad’. Zeaxanthin content was very similar in all the studied samples. Lutein was not determined in the *R. gallica* samples. When examining the total content of carotenoids, no statistically significant difference between *R. gallica* and its cultivars was observed. The content was lowest in ‘Splendens’ and highest in ‘Violacea’. In addition, when comparing *R. spinosissima* and its cultivar, no statistically significant difference in the total carotenoid content was perceived. The highest content was found in ‘Fruhligsmorgen’ and the lowest in ‘Fruhlingsduft’. There is also no statistically significant difference between *R. pendulina* and its cultivars.

From Figure 3, it can be observed that the samples are divided into five groups based on their total carotenoid content. The first includes ‘Mount Everest’ (*R. pendulina* group), ‘Fruhlingsduft’ (*R. spinosissima* group), and *R. spinosissima*. The ‘Harstad’ (*R. pendulina* group), ‘Splendens’ (*R. gallica* group), and ‘Bourgogne’ (*R. pendulina* group) cultivars, and *R. gallica* form the second group. The third group includes ‘Fruhlingsmorgen’ (*R. spinosissima* group) and ‘Single cheery’ (*R. spinosissima* group) represents the fourth group. The last, fifth group includes *R. pendulina*, ‘Poppius’ (*R. spinosissima* group), and ‘Violacea’ (*R. gallica* group). Again, very few cultivars were placed in the same group as the main species from which they are derived.

## 3. Discussion

Based on the research, the content of specific bioactive compounds in the rose hips of three native Slovenian rose species and various cultivars derived from them, growing under the same conditions in the Arboretum Volčji Potok (Slovenia), was determined.

Such a similar comparison has already been made by Jayram [36]. He concluded that the ascorbic acid content does not depend on the time of the cultivar origin. He argued that the modern cultivar, ‘Shining Ruby’, crossed in 1992, had the highest ascorbic acid levels compared to the old cultivar, ‘Blush’, crossed in 1752. Comparing the above results with the results of this study, it can be observed that the results are not in agreement with Jayram’s findings [36]. When comparing within individual groups, the modern cultivars (crossed after 1876) analyzed in this study have lower ascorbic acid levels than the old cultivars and original genotypes. However, when comparing all the analyzed samples, ‘Mount Everest’ stands out with higher values (7.11 g/kg FW) compared to some old cultivars and original genotypes. It is assumed that these values originate from *R. pendulina*, which already has a rather high ascorbic acid content (12.36 g/kg FW) and that the ‘Mount Everest’ cultivar is derived from this species. Tabaszewskaa and Najgebauer-Lejko [37] determined the content of ascorbic acid in the samples of *R. canina*. They found that 1 L of tincture from fresh rose hips contains 0.3 g of ascorbic acid. Fascella et al. [38] reported the ascorbic acid content of the samples belonging to *R. canina*, *R. corymbifera*, *R. micrantha*, and *R. sempervirens*. The contents varied from 2.2 g/kg DM (*R. corymbifera*) to 5.13 g/kg DM (*R. canina*). The results for ascorbic acid ranged from 1.24 g/kg to 12.79 g/kg FW in the samples of this study and considering that the DW of these samples represents approximately 37% of FW (63% water in samples), it is noted that this study’s contents ranged from 0.46 g/kg DW (‘Single Cherry’) to 4.71 g/kg DW (‘Harstad’). Demir et al. [28] reported the ascorbic acid content of samples from five different *Rosa* species. The content ranged from 0.66 g/kg DW to 1.60 g/kg DW. Roman et al. [39] reported ascorbic acid in *R. canina* ranging from 1.12 g/kg to 3.60 g/kg of frozen pulp. All these results demonstrate that this study’s rose hips are rich in ascorbic acid. Skrypnik et al. [40] studied the rose hips of *R. canina* and *R. rugosa* collected from wild shrubs in the Kaliningrad region, Russia. They found that *R. rugosa* has an extremely high ascorbic acid content compared to *R. canina*. Values of approximately 10 g/kg body weight were reported for *R. canina*, and as high as 40 g/kg body weight for *R. rugosa*, which is 10 times higher than the values determined in this study and those of other studies used for comparison. Such great differences are probably because the samples examined in the study by Skrypnik et al. [40] grew under completely different environmental conditions than in this study, resulting in a higher accumulation of ascorbic acid. An important factor is certainly the influence of genotype as well, as explained by Skrypnik et al. [40].

The organic acids determined in this paper’s study were citric, malic, quinic, shikimic, and fumaric. Many of the highest levels of total organic acids were once more found in the cultivars crossed before 1876. Of all the samples analyzed, *R. spinosissima* had the lowest content (1.57 g/kg FW), while ‘Harstad’ (34.39 g/kg FW), an older cross of *R. pendulina*, had the highest. Among the modern cultivars, ‘Fruhlingsmorgen’ and ‘Poppius’ stood out, showing high contents (25.46 g/kg and 23.93 g/kg FW) compared to the original *R. spinosissima* genotype (1.57 g/kg FW). The two cultivars, ‘Fruhlingsmorgen’ and ‘Poppius’, are known to be derived from the *R. spinosissima* genotype and were crossed in 1872 by the breeder, Stenberg (‘Poppius’), and in 1941, by the breeder, Kordes (‘Fruhlingsmorgen’). *R. spinosissima* was intensively used for breeding in Canada due to its extremely good winter hardiness and low water requirement, which can be a possible reason for the high content of organic acids in this study’s examined fruits. The environmental conditions at the place and time of cultivation are also of great importance for the results of the organic acid content. Comparing the results of this study with the results of a previously published study [14] for *R. pendulina*, it can be noticed that the total organic acid content of rose hips autochthonously grown in the Čaven area (Slovenia) was 2.92 g/kg FW, which was significantly lower than the total organic acid content of the examined samples from the Arboretum Volčji Potok. Based on the 37% share represented by DW with respect to FW in these samples, it was found that the total organic acid content ranged from 0.58 g/kg DW (*R. spinosissima*) to 12.72 g/kg DW (‘Harstad’), which is significantly more than the data reported by Cunja et al. [27], who found that the total organic acid content of *R. canina* harvested on six different dates ranged from 0.26 g/kg DW to 0.36 g/kg DW. The organic acid with the highest content in the study by Cunja et al. [27] was shikimic; however, it was the least abundant in this study’s samples. In comparison, the most abundant acid was quinic acid. What is more, the results of the study by Kunc et al. [14] showed that the content of shikimic acid in *R. pendulina* was the lowest of all the acids analyzed. Cunja et al. [27] also determined a lower level of tartaric acid, which was not detected in this study’s samples. Adamczak et al. [41] reported that the amount of citric acid in the samples of sect. Caninae was 0.03 g/kg DW, which is significantly lower than the lowest level detected in this study for *R. spinosissima* (0.16 g/kg DW). Demir et al. [28] reported the organic acid content of five rose hips, including *R. gallica*, which contained up to 47.6 g/kg DW of citric acid, that is more than the total organic acid content in this study. The reason for such a high content is presumably that the roses from that study grew in the wild in Turkey where they were exposed to greater stress factors than the samples analyzed for this study, which grew in a maintained park with optimal growth conditions (temperature, watering, nutrients…).

Based on the results obtained by analyzing all five carotenoids (lutein, zeaxanthin, lycopene, *α*-carotene, and *β*-carotene) in rose hips, it was observed that the lowest value (8.96 mg/100 g FW, ‘Spring Fragrance’) and the highest value (100.84 mg/100 g FW, ‘Spring Morning’) were determined in modern cultivars, the *R. spinosissima*-derived Scot varieties. As aforementioned, the ‘Fruhlingsmorgen’ cultivar, crossed in Canada, is better adapted to winter conditions, which could be due to different weather conditions in Slovenia and Canada, resulting in the cultivar being exposed to higher temperatures and stronger UV radiation, namely stress; consequently, the cultivar produces carotenoids in greater amounts. Alp et al. [42] reported that the total carotenoid content of *R. dumalis* ranged from 47 to 85 mg/100 g FW. When the obtained results are converted to DW, based on the ratio of 37% DW compared to FW, it can be observed that the total carotenoid content of this study’s analyzed samples ranged from 3.32 mg/100 g DW (‘Fruhlingsduft’) to 37.31 mg/100 g DW (‘Fruhlingsmorgen’), which was less than reported. Such lower levels in this study are most likely a result of the well-maintained rose bushes in the park from which the samples were obtained, and these conditions represented a lower exposure for one plant. Compared to the rose hip samples of *R. gallica* from a cultivated park analyzed in this experiment, it is observed that the rose hips of *R. gallica* from nature reported by Kunc et al. [43] had a higher carotenoid content (32.39 mg/100 g FW) in the same year. Kunc et al. [43] also concluded that the reported carotenoid levels are at a lower limit compared to those reported in the literature and that there is still potential for higher carotenoid levels. Such differences occur due to genetic variability, agrometeorological conditions, growing conditions, storage, and differences in maturity, in addition to technological differences [33]. Cunja et al. [27] determined that the *β*-carotene content in *R. canina* ranged from 6.5 mg/100 g DW to 22.1 mg/100 g DW at six different harvest dates. Medveckiene et al. [44] found that the content of *β*-carotene ranged from 3.95 mg/100 g to 31.4 mg/100 g DW. In the samples of this study, it ranged from 2.60 mg/100 g DW (‘Fruhlingsduft’) to 36.58 mg/100 g (‘Fruhlingsmorgen’), and the lycopene content ranged from 0.02 mg/100 g FW to 4.11 mg/100 g FW. Turkben et al. [45] reported the lycopene content in *R. canina* of approximately 10.39 mg/100 g FW. According to Baranski et al. [46], the reason for the low lycopene content and accumulation of *β*-carotene could be a plant’s response to sunburn. Furthermore, Rodriguez-Amaya et al. [47] reported that increased temperature and sunlight exposure can increase carotene formation in fruits and promote carotenoid degradation. They found that fruits of the same acerola, mango, and papaya varieties, grown in hot regions, contained significantly higher concentrations of carotenoids than fruits grown in temperate climates. Leafy vegetables grown in greenhouses or on plots covered with plastic mulch have higher concentrations of carotenoids in summer. In contrast, the carotenoid content of leafy vegetables grown in open fields is significantly lower in summer, indicating that photodegradation is given priority over enhanced carotenogenesis.

## 4. Materials and Methods

### 4.1. Plant Material

Fruits of 12 rose hips collected from the Arboretum Volčji Potok, Slovenia, at the stage of full ripeness, BBCH 88 [1], were included in this study. The rose hips examined were the three main rose hip species: *R. gallica*, *R. spinosissima*, and *R. pendulina*, and rose hips of cultivars derived from these species: *R. gallica* ‘Violacea’, *R. gallica* ‘Splendens’, *R. spinosissima* ‘Poppius’, *R. spinosissima* ‘Fruhlingsduft’, *R. spinosissima* ‘Single Cherry’, *R. spinosissima* ‘Fruhlingsmorgen’, *R. pendulina* ‘Harstad’, *R. pendulina* ‘Bourgogne’, and *R. pendulina* ‘Mount Everest’. *R. spinosissima* and its cultivars are known as Scotch roses, and older *R. spinosissima* cultivars as Burnet roses (Table 4) [48].

The ‘Violacea’ cultivar was crossed in the Netherlands in 1795, and its breeder is unknown. ‘Splendens,’ crossed in 1583, also has an unknown breeder. ‘Poppius’ was crossed in Sweden, at Stenberg, in 1872. ‘Fruhlingsduft’ and ‘Fruhlingsmorgen’ were crossed at Kordes, Germany, in 1941. ‘Single Cherry’ was crossed before 1962 (breeder unknown). ‘Harstad’ is an older cultivar from an unknown breeder. ‘Mount Everest’ was crossed in 1956 and ‘Bourgogne’ in 1983 in the Netherlands by Interplant (Table 4).

All plants from which rose hips were collected grew under the same climatic conditions (Figure 4). The collected material was placed on ice and brought to the laboratory. Analysis of ascorbic acid was performed immediately on the fresh rose hips. The material for other analyses was stored at −20 °C until further examination.

The Arboretum Volčji Potok is located in central Slovenia, at 304 m above sea level. It houses approximately 3500 species and varieties of coniferous and deciduous trees, shrubs, and wild perennials. The trees and shrubs are basic botanical species from Europe, North America, and Asia, as well as cultivars grown for ornamental purposes. Among the deciduous trees in the arboretum, maple, linden, birch, and beech are the most represented species. In addition, roses also thrive, arranged in three neat rose crowns. All roses are cared for, pruned, fertilized, and watered by the Arboretum’s caretakers throughout the year (Figure 5).

### 4.2. Extraction and Analysis of Ascorbic Acid

The ascorbic acid extraction on the rose hips (flesh with the skin, without seeds) was carried out as previously described by Kunc et al. [49]. An amount of 15 mL of 3% *meta*-phosphoric acid was added to 0.5 g of material. Firstly, the samples were left at room temperature for 30 min on an orbital shaker platform (Unimax 1010, Heidolph Instruments, Schwabach, Germany), then they were centrifuged with Eppendorf 5810 R Centrifuge (Hamburg, Germany) at 10,000× *g* for 5 min at 4 °C and filtered through a Chromafil A-20/25 cellulose mixed ester filter (Macherey-Nagel, Düren, Germany) into vials. The vials with extracts were stored at −20 °C until further examination. The samples were analyzed using an HPLC system (Vanquish UHPLC, Thermo Fisher Scientific) and a UV detector set at 245 nm. The chromatographic conditions for ascorbic acid determination were the same as reported by Mikulic-Petkovsek et al. [50]. For the separation of ascorbic acid, a Rezex ROA column (Phenomenex) operated at 20 °C with 4 mM sulfuric acid for the mobile phase was used.

### 4.3. Extraction and Analysis of Organic Acids

The seeds were first separated from the flesh with skin and then discarded. Sample extraction for organic acid quantification was carried out following the protocol reported by Kunc et al. [14]. The flesh with skin was finely chopped. An amount of 1 g of material was weighed into centrifuge tubes, and 3 mL of bi-distilled water was added. The tubes were placed on a shaker (Unimax 1010, Heidolph Instruments, Schwabach, Germany) for half an hour. The extract was centrifuged with Eppendorf centrifuge 5810 R at 10,000× *g* for 7 min at 4 °C. The supernatant was filtered through a syringe filter (Chromafil Xtra MV-20/25, Macherey Nagel, Düren, Germany) into a vial, labeled, and stored at −20 °C. The extraction was performed in triplicate. A UV detector was used for organic acid separation, set at 210 nm, a Rezex ROA column (Phenomenex, Torrance, CA, USA), and heated to 65 °C; the mobile phase was 4 mM sulfuric acid with a flow rate of 0.6 mL/min.

### 4.4. Extraction and Analysis of Carotenoids

The extraction of carotenoids was carried out according to the method previously described by Mikulic-Petkovsek et al. [51]. An amount of 0.2 g of frozen material (flesh with skin, without seeds) was extracted in glass centrifuges with 2 mL of acetone at a temperature of 4 °C with an Ultra-Turrax homogenizer for 30 s and identified on an Accela HPLC system (Thermo Scientific, San Jose, CA, USA) via a gradient method. The samples were then filtered through Chromafil A-20/25 polyamide/nylon filters (Macherey-Nagel, Düren, Germany) into labeled vials. The extracts were immediately analyzed on HPLC-DAD (Thermo Finnigan, San Jose, CA, USA) and a Gemini C18 column (Phenomenex). The 1st mobile phase was solvent A: acetonitrile/methanol/water (100/10/5, *v*/*v*/*v*) and the 2nd mobile phase was solvent B: acetone/ethyl acetate (2/1, *v*/*v*). The flow rate was 1 mL/min and the gradient method employed was reported by Mikulic-Petkovsek et al. [51].

### 4.5. Statistical Analysis

For statistical analysis, data were collected using Microsoft Excel 2016 and R commander. One-way analysis of variance (ANOVA) was used. Tukey’s test was used to compare treatments when ANOVA showed significant differences between values. Results were given as mean ± standard error (SE) of fresh weight (FW). If the *p*-values were lower than 0.05, the differences between the genotypes were statistically significant. Individual results were shown in tables and hierarchical cluster analysis (cluster dendrogram according to Ward’s classification method) was used to compare the total content of bioactive compounds among all the analyzed samples.

## 5. Conclusions

Bioactive compounds in the rose hips of three autochthonous species (*R. pendulina*, *R. spinosissima*, and *R. gallica*) and nine cultivars derived from these species (‘Violacea’, ‘Splendens’, ‘Poppius’, ‘Fruhlingsmorgen’, ‘Fruhlingsduft’, ‘Sigle Cherry’, ‘Harstad’, ‘Bourgogne’, and ‘Mount Everest’) were studied.

In general, it can be concluded that the content of bioactive compounds in modern cultivars is lower than in old cultivars and original rose hip species included in this study, with some exceptions. The results confirmed that modern breeding is mainly focused on goals, such as beautiful flowering, fragrant flowers, rapid repeat flowering, and resistance to diseases and pests, and not on the content of bioactive compounds. Given that the importance of functional foods and the wealth of the Rosa genus with bioactive compounds is increasing in the foreground, an additional goal of breeding in the future could be the introduction of varieties intended primarily for the production of high-quality rose hips. Such modern breeding will be an important source of bioactive compounds needed by humans for normal functioning and will contribute to a wide variety of rose cultivars. The importance of roses will also be shifted to the field of functional foods as a rich source of bioactive components.

## Figures and Tables

**Figure 1 plants-12-03734-f001:**
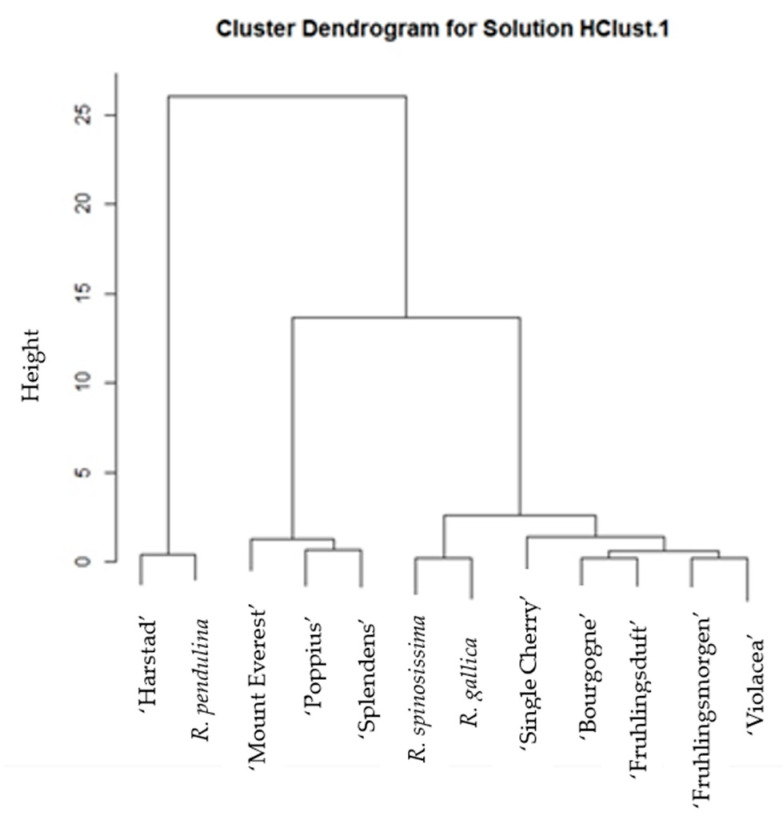
Hierarchical cluster analysis of the content of ascorbic acid in flesh with skin for all 12 analyzed rose hips.

**Figure 2 plants-12-03734-f002:**
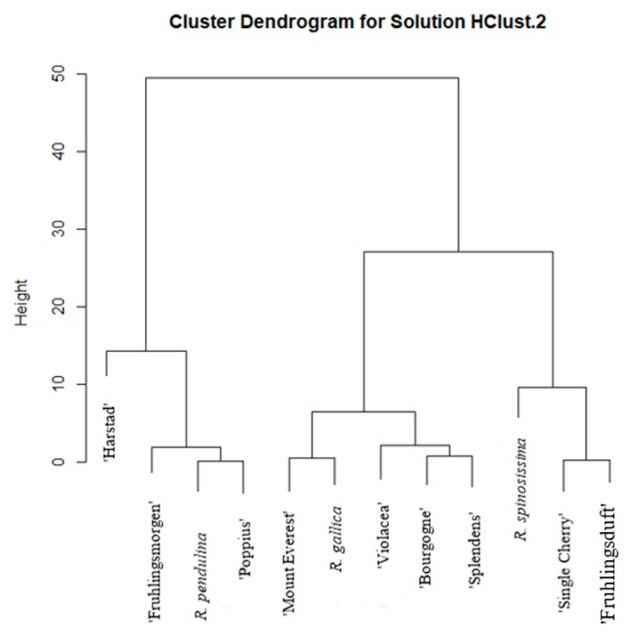
Hierarchical cluster analysis of the total content of organic acids in flesh with skin for all 12 analyzed rose hips.

**Figure 3 plants-12-03734-f003:**
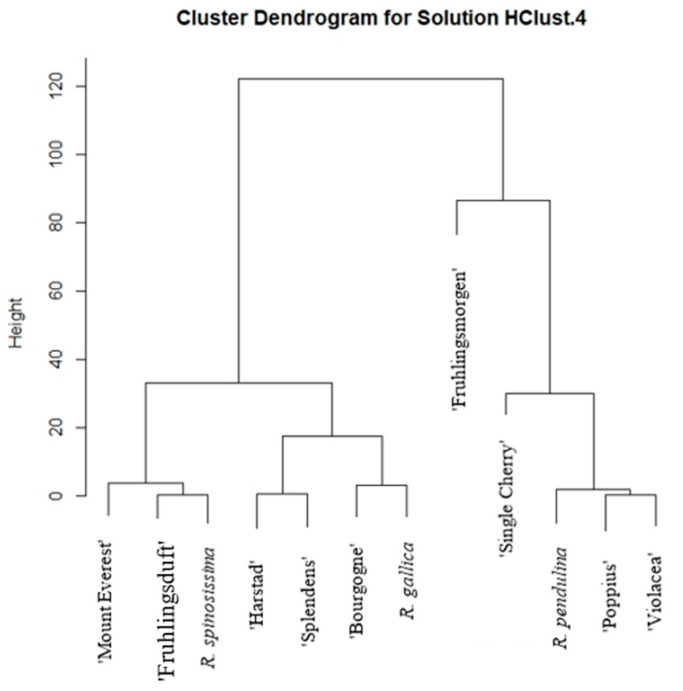
Hierarchical cluster analysis of the total content of carotenoids in flesh with skin for all 12 analyzed rose hips.

**Figure 4 plants-12-03734-f004:**
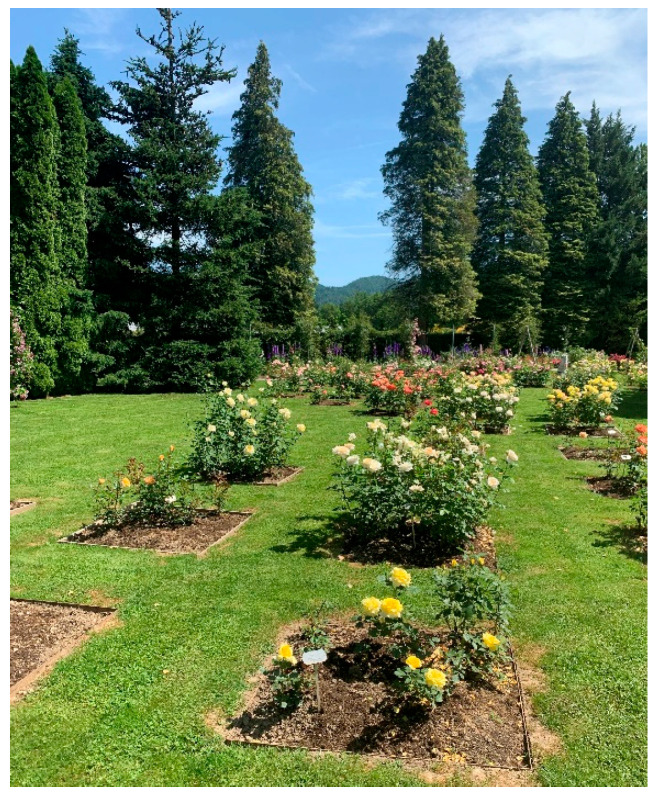
Rosary Arboretum Volčji Potok.

**Figure 5 plants-12-03734-f005:**
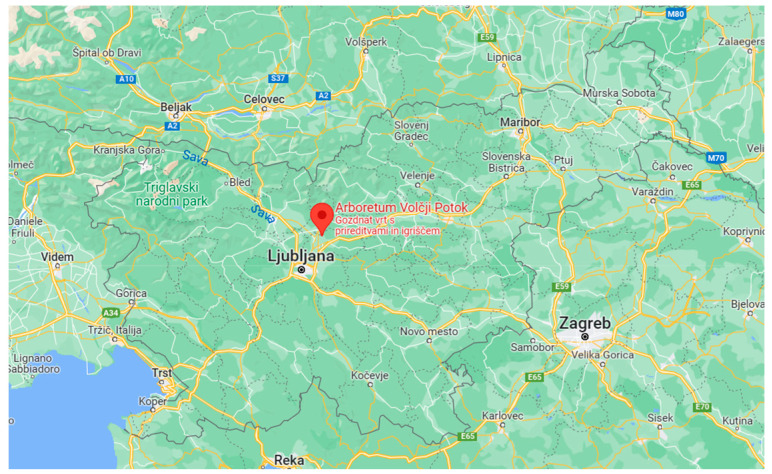
Location of the Arboretum Volčji Potok.

**Table 1 plants-12-03734-t001:** The average content of fresh weight (FW) ± standard error (g/kg FW ± SE) of ascorbic acid in rose hips of analyzed rose hips. Different letters indicate a statistical difference within each genotype and their cultivars.

Species and Cultivars	Content
*R. gallica*	3.16 ± 0.32 ^a^
‘Violacea’	2.17 ± 0.49 ^a^
‘Splendens’	6.32 ± 0.77 ^b^
*R. spinosissima*	3.39 ± 0.44 ^bc^
‘Poppius’	5.67 ± 0.49 ^c^
‘Fruhlingsduft’	2.57 ± 0.27 ^ab^
‘Single cherry’	1.24 ± 0.66 ^a^
‘Fruhlingsmorgen’	1.93 ± 0.16 ^a^
*R. pendulina*	12.36 ± 0.77 ^c^
‘Harstad’	12.79 ± 0.44 ^c^
‘Bourgogne’	2.38 ± 0.13 ^a^
‘Mount Everest’	7.11 ± 0.47 ^b^

**Table 2 plants-12-03734-t002:** The average content ± standard error (g/kg FW ± SE) of organic acids (citric, malic, quinic, shikimic, and fumaric acid) in rose hips of the analyzed rose hips. Different letters indicate a statistical difference within each genotype and their cultivars.

Species and Cultivars	Citric Acid	Malic Acid	Quinic Acid	Shikimic Acid	Fumaric Acid	Total
*R. gallica*	5.14 ± 1.41 ^b^	2.05 ± 0.85 ^ab^	10.87 ± 4.50 ^a^	0.05 ± 0.01 ^b^	0.01 ± 0.001 ^a^	18.12 ± 6.77 ^a^
‘Violacea’	3.68 ± 1.23 ^a^	1.49 ± 0.63 ^a^	8.59 ± 3.01 ^a^	0.02 ± 0.008 ^a^	0.008 ± 0.004 ^b^	13.79 ± 4.88 ^a^
‘Splendens’	2.69 ± 0.99 ^a^	3.28 ± 1.58 ^b^	9.28 ± 3.42 ^a^	0.04 ± 0.007 ^ab^	0.01 ± 0.002 ^ab^	15.3 ± 5.99 ^a^
*R. spinosissima*	0.42 ± 0.22 ^a^	0.25 ± 0.11 ^a^	0.89 ± 0.34 ^a^	0.01 ± 0.004 ^a^	0.002 ± 0.001 ^a^	1.57 ± 0.68 ^a^
‘Poppius’	4.18 ± 1.75 ^b^	3.34 ± 1.19 ^b^	16.36 ± 8.29 ^c^	0.04 ± 0.01 ^b^	0.008 ± 0.002 ^b^	23.93 ± 11.24 ^a^
‘Fruhlingsduft’	4.23 ± 1.19 ^b^	3.24 ± 1.27 ^b^	4.14 ± 2.09 ^ab^	0.11 ± 0.06 ^c^	0.007 ± 0.002 ^b^	8.73 ± 4.61 ^a^
‘Single cherry’	2.95 ± 1.17 ^ab^	1.52 ± 0.56 ^ab^	4.57 ± 1.62 ^ab^	0.05 ± 0.03 ^b^	0.006 ± 0.004 ^ab^	9.10 ± 3.38 ^a^
‘Fruhlingsmorgen’	12.09 ±5.23 ^c^	6.89 ± 2.17 ^c^	6.29 ± 3.10 ^b^	0.18 ± 0.05 ^d^	0.006 ± 0.001 ^ab^	25.46 ± 10.55 ^a^
*R. pendulina*	1.67 ± 0.46 ^a^	0.66 ± 0.38 ^a^	21.68 ± 13.53 ^b^	0.05 ± 0.02 ^b^	0.003 ± 0.001 ^a^	24.06 ± 14.39 ^a^
‘Harstad’	4.47 ± 2.09 ^c^	3.31 ± 1.36 ^c^	26.52 ± 12.97 ^b^	0.08 ± 0.02 ^c^	0.008 ± 0.003 ^b^	34.39 ± 16.44 ^a^
‘Bourgogne’	3.75 ± 1.20 ^c^	2.41 ± 0.92 ^bc^	9.86 ± 3.46 ^a^	0.04 ± 0.03 ^ab^	0.006 ± 0.001 ^b^	16.07 ± 5.61 ^a^
‘Mount Everest’	2.71 ± 1.29 ^b^	1.75 ± 0.96 ^ab^	11.43 ± 6.58 ^a^	0.03 ± 0.01 ^a^	0.006 ± 0.003 ^b^	18.64 ± 8.87 ^a^

**Table 3 plants-12-03734-t003:** The average content ± standard error (mg/100 g FW ± SE) of carotenoids (lutein, zeaxanthin, lycopene, *α*-carotene, *β*-carotene) in rose hips of the analyzed rose hips. Different letters indicate a statistical difference within each genotype and their cultivars.

Species and Cultivars	Lutein	Zeaxanthin	Lycopene	*α*-Carotene	*β*-Carotene	Total
*R. gallica*	-	0.005 ± 0.004 ^a^	0.66 ± 0.35 ^ab^	0.16 ± 0.09 ^a^	25.25 ± 11.73 ^a^	26.25 ± 12.17 ^a^
‘Violacea’	0.88 ± 0.26 ^b^	0.03 ± 0.01 ^a^	0.24 ± 0.15 ^a^	0.20 ± 0.09 ^a^	36.88 ± 13.09 ^a^	38.23 ± 13.60 ^a^
‘Splendens’	0.55 ± 0.12 ^a^	0.02 ± 0.01 ^a^	0.15 ± 0.05 ^b^	0.18 ± 0.04 ^a^	17.00 ± 6.61 ^a^	17.90 ± 6.83 ^a^
*R. spinosissima*	0.99 ± 0.11 ^a^	0.05 ± 0.02 ^b^	0.02 ± 0.01 ^a^	-	8.20 ± 2.04 ^a^	9.26 ± 2.18 ^a^
‘Poppius’	1.23 ± 0.07 ^a^	0.05 ± 0.02 ^b^	0.07 ± 0.04 ^ab^	0.37 ± 0.10 ^b^	36.34 ± 12.16 ^b^	38.06 ± 12.39 ^a^
‘Fruhlingsduft’	1.56 ± 0.14 ^a^	0.01 ± 0.005 ^a^	0.03 ± 0.02 ^a^	0.32 ± 0.1^6 a^	7.04 ± 2.16 ^a^	8.96 ± 2.49 ^a^
‘Single cherry’	0.24 ± 0.11 ^a^	-	0.13 ± 0.03 ^b^	-	57.69 ± 21.45 ^b^	58.06 ± 21.59 ^a^
‘Fruhlingsmorgen’	1.55 ± 0.71 ^a^	0.008 ± 0.003 ^a^	0.08 ± 0.04 ^ab^	0.33 ± 0.18 ^a^	98.87 ± 39.34 ^c^	100.84 ± 40.27 ^a^
*R. pendulina*	0.84 ± 0.32 ^a^	0.04 ± 0.02 ^a^	4.07 ± 1.77 ^b^	6.14 ± 2.02 ^b^	25.68 ± 13.06 ^b^	36.77 ± 17.19 ^a^
‘Harstad’	0.66 ± 0.21 ^a^	0.02 ± 0.01 ^a^	4.11 ± 1.64 ^b^	4.16 ± 1.85 ^b^	9.52 ± 4.31 ^a^	18.47 ± 8.02 ^a^
‘Bourgogne’	0.66 ± 0.15 ^a^	0.05 ± 0.02 ^a^	2.35 ± 0.72 ^b^	3.86 ± 1.43 ^b^	22.34 ± 11.04 ^b^	29.26 ± 13.36 ^a^
‘Mount Everest’	2.57 ± 0.32 ^b^	0.04 ± 0.03 ^a^	0.11 ± 0.05 ^a^	0.13 ± 0.07 ^a^	9.06 ± 3.78 ^a^	11.91 ± 4.25 ^a^

Note: (-) Compound was not detected.

**Table 4 plants-12-03734-t004:** Origin data for cultivars used in this study.

Cultivar	Species of the Origin	Breeding Company	Year of Origin
‘Violacea’	*R. gallica*	unknown (The Netherlands)	1795
‘Splendens’	*R. gallica*	unknown	1583
‘Poppius’	*R. spinosissima*	Stenberg (Sweden)	1872
‘Fruhlingsduft’	*R. spinosissima*	Kordes (Germany)	1941
‘Fruhlingsmorgen’	*R. spinosissima*	Kordes (Germany)	1941
‘Single cherry’	*R. spinosissima*	unknown	1962
‘Harstad’	*R. pendulina*	unknown	unknown (old variety)
‘Mount Everest’	*R. pendulina*	Interplant (The Netherlands)	1956
‘Bourgogne’	*R. pendulina*	Interplant (The Netherlands)	1983

## Data Availability

Data will be made available on request.

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
