# Peer review of "Breeding of Modern Rose Cultivars Decreases the Content of Important Biochemical Compounds in Rose Hips"

_plants, 2023, doi:10.3390/plants12213734_

Round 1
Reviewer 1 Report
Comments and Suggestions for Authors
The topic of this article is clear, rational, rich content and strong innovation. However, some minor issues still need to be improved. After minor revision, it can be accepted.
1) The author should check the whole manuscript for any grammatical errors and any other differences. Writing needs considerable improvement.
2) Line 83, Line 116 and Line 119, change “et al.” to “et al.”. Please check this manuscript to avoid similar errors.
3) Please pay attention to the “a,b,c” notation in the table, which should be set to superscript.

Extensive editing of English language required
Reviewer 2 Report
Comments and Suggestions for Authors
The manuscript entitled "Breeding of modern rose cultivars decreases the content of important biochemical compounds in rose hips", from the authors Nina Kunc, Metka Hudina, Maja Mikulic-Petkovsek and Gregor Osterc.
The manuscript is interesting, well conceived, the experimental results are clearly presented and analyzed. However, the results are only a statement that decreases the content of important biochemical compounds in rose hips by breeding of modern rose cultivars. Although the scientific result is relatively small, it can be a contribution to the study in this field.
In tables 1, 2 and 3, the meaning of labels a, b, c, ab and d is not defined. I ask the authors to clarify these marks.
I consider that manuscript should be published in journal „Plants“ after minor corrections.
Reviewer 3 Report
Comments and Suggestions for Authors
The results presented in this work are quite interesting and therefore I believe that deserves to be published in Plants. In this paper the nature and relative amounts of secondary metabolites in plants obtained by crossing of native species has been determined. Specifically, the content of ascorbic acid, organic acids and carotenoids has been determined.
However, it is well known that the content of secondary metabolites depends on the growing conditions of plants. So, I think that authors should emphasize in the introduction that this study was carried with different species of rose hips that were grown under the same conditions. This concept should be used in the discussion as well. It is not possible to compare results from measurements made with 100 years difference. I think that the whole discussion should be rewritten considering this fact. No valid conclusion on effect of breeding can be obtained by comparing metabolites content of different species grown and studied under different conditions.
In addition, the following points should be addressed before the work is accepted for publication.
Line 42. The effect of vitamin C is not “to purify ROS”. This must be changed to “eliminat Ros” or “get rid of ROS”
Line 43-45. This phrase should be changed to “It has been shown demonstrated that ascorbic acid is essential for plant growth” Ref.
Line 49. Changes “ It plays an important role …” by “On the other hand, ascorbic acid plays an important role”
Line 52. Delete “At the human population level” and starts this phrase with “In addition, “ or “Additionally, “
Line 76. Replace “They represent” with “They are”
Line 83. This reference must be changed. Hypertension as health problem is a very important issue and surely there are many specialized reports on it.
Line 110-116. This paragraph describes the effect of carotenoids on human health. References must be added. Ref 33 is not related to what is established here.
Line 140. The meaning of abbreviations should be explained. “FW”
Line 141. The value of R. penduline is 12.36
Comments on the Quality of English LanguageExtensive revision of english should be made
Reviewer 4 Report
Comments and Suggestions for Authors
The authors give an extensive review in the introduction on the benefits of the compounds that can be found in rose hips but this manuscript is about how breeding reduces the amount of bioactive compounds in rose hips. It would be more important to give examples from the literature of this same observation either in other roses, flowers, or other plants.
That being said, I do not agree with the observation that bioactive compounds in rose hip cultivars decreases compared to the original species based on the data presented in the manuscript:
-In Table 1 there is one cultivar that contains more ascorbic acid that the original species of each group.
-In Table 2 all the cultivars have more of all the organic acids present than the original species (R. spinosissima) and ‘Harstad’ has more of all the organic acids present than the original species (R. pendulina).
-In Table 3, ‘Fruhlingsmorgen’ has more all the carotenoids present than the original species (R. spinosissima)
What would be interesting is to compare the amount of bioactive compounds in both parents of a cross to the progeny not just a single parent. This could tell us more about how bioactive compounds are transferred through breeding.
The authors need to address the value of roses as ornamentals versus their value for food use. Can’t these compounds be obtained from consumption of other foods? Why would we start consuming rose hips?
Figure 1.: The quality of this image needs to be improved. It is difficult to read the cultivar names.
Also, not all readers are familiar with Cluster Dendrograms. The authors need to communicate what the dendrogram illustrates. How does a Cluster Dendrogram help visualize the data? The authors state the cultivars are clustered into three groups depending on the amount of ascorbic acid present and that very few cultivars are classified in the same group as the main species, but what does this mean?
Comments on the Quality of English LanguagePlease review for minor grammatical or spelling errors.
Reviewer 5 Report
Comments and Suggestions for Authors
In this article, the authors compared the content and composition of bioactive compounds in autochthonous rose hips with the content of bioactive compounds in some cultivars. The article is very interesting and I would like to ask the authors to complete their analyzes by performing an assay of total polyphenols and total antioxidant activity. Furthermore, I ask the authors to limit as much as possible any reference to the health properties of the molecules studied, since the present work is mainly addressed to the botany and chemistry of plants. I therefore advise the authors, if they want to delve deeper into the health aspect of plants, to collaborate with specific research groups.
Comments on the Quality of English LanguageMinor editing of English language required
Round 2
Reviewer 3 Report
Comments and Suggestions for Authors
In the revised version authros have addressed all miy oiints and suggestions. Thus, I think it could be published in its current form
Reviewer 4 Report
Comments and Suggestions for Authors
No comments or suggestions.
Reviewer 5 Report
Comments and Suggestions for Authors
I thank the authors for answering my questions